# The Contribution of the User Experiences Goals for Designing Better Cobots: A Systematic Literature Review

Inês Margarida Duarte [1,*], Ana Pinto [2], Carla Carvalho [3], Ana Zornoza [4] and Joana Santos [5,6]

1   University of Coimbra, Faculty of Psychology and Educational Sciences, 3000-115 Coimbra, Portugal
2   Centre for Business and Economics Research (CEBER), Faculty of Sciences and Technology,
    University of Coimbra, 3030-790 Coimbra, Portugal
3   Center for Research in Neuropsychology and Cognitive Behavioral Intervention (CINEICC),
    Faculty of Psychology and Educational Sciences, University of Coimbra, 3000-115 Coimbra, Portugal
4   Faculty of Psychology, University of Valencia, 46010 Valencia, Spain
5   Polytechnic Institute of Porto, School of Health (ESS | P.PORTO), Scientific Area of Environmental Health,
    4200-072 Porto, Portugal
6   Associate Laboratory of Energy, Transports and Aerospace (LAETA)/Institute of Science and Innovation in
    Mechanical and Industrial Engineering (INEGI), Faculty of Engineering, University of Porto,
    4200-462 Porto, Portugal
*   Correspondence: inesmargaridalduarte@gmail.com; Tel.: +351-925-908-531

**Abstract:** Collaborative robots are an indispensable element of both industry 4.0 and industry 5.0, the latter of which gives special emphasis to the human facet of the human-robot collaboration. To facilitate such an interaction, attention should be given to the design of the cobot, including its interface, which enables communication with the user. Programming through the interface and performing a task with the robotic device are responsible for the user experience (UX), which comprises both pragmatic and hedonic aspects. In order to design the most positive experience for users, their perspectives must be considered, which is achieved through the identification of UX goals. In this respect, a systematic review was conducted to revise the UX goals present in the literature. The following seven UX goals were identified: safety, relationship, usability, inspiration, flexibility, efficiency, and accomplishment. These findings represent the first systematic categorization of UX goals for the specific design of cobots, that should empirically be tested.

**Keywords:** collaborative robot; user experience goals; experience-driven design; systematic literature review

## 1. Introduction

In the context of the fourth industrial revolution, robots are being increasingly used in the industrial workforce. Industry 4.0 is characterized by mass production [1] and mass customization [2]. The target is smart manufacturing with high rates of productivity, achieved through different innovative technologies, namely robotics and artificial intelligence [3], that complement humans' capacities. The presence of humans is therefore still recognized as necessary for achieving the required customization in manufacturing, as they take responsibility for the tasks that require higher levels of cognition [4].

Notwithstanding, the role of human workers is further enhanced considering the emergent industry 5.0, whose core target is to achieve mass personalization [2]. According to Hanif and Iftikhar [1], contrary to the previous four phases, which stepped into dehumanization, this fifth industrial revolution emphasizes how technology should be used for the benefit of individuals, by focusing on the personalized demands and requirements of customers [2]. To achieve that, Demir and colleagues [3] suggest humans shall co-work with the robotic machines in all possible situations and contexts, through the vast integration of robots in organizations.

Despite the controversy revolving around whether this fifth revolution has started yet [1], both industry 4.0 [4] and industry 5.0 [2] highlight human-robot collaboration (HRC) as a key aspect when pursuing the fulfillment of their respective objectives. The manufacturing should be human-centered, namely, the well-being of the workers has to be placed at the centre of the process [5,6].

HRC can be identified as the third level of human-robot interaction [4]. In addition to coexistence (i.e., common workspace and time) and cooperation (i.e., common workspace, time, aim, and resources), the collaboration of a robot and a human implies the existence of direct physical contact between them e.g., [7]. For that, user interfaces (UIs) are of extreme importance, as they are the main channel of communication connecting the two mentioned entities [4] and contribute to its efficiency and efficacy [8].

Given the complexity of human-robot interaction, an interdisciplinary approach is beneficial, including inputs both from the engineering and the psychological fields of knowledge [9]. An example of a robotic system encompassed by such an approach, present in both industry 4.0 [10] and industry 5.0 [1], is one of collaborative robots (or cobots). A cobot is, by definition, a "robot designed for direct interaction with a human within a defined collaborative workspace" [11]. This kind of robot is being adopted at unprecedented rates in organizations and it is expected to become the central tool of manufacturing globally, due to its specific characteristics, such as safe interaction with humans [12].

The adoption of cobots has been growing in manufacturing since they offer an opportunity for the human and robot to exchange information and share tasks. In order to improve this interactive experience with benefit for both, it is necessary that the robot understand the human, and the human to understand the robot [13]. This social dimension approach was introduced by Industry 5.0 which complements the existing Industry 4.0 approach. Therefore, in this context it is expected that innovation and research will be positioned at the service of the transition to a sustainable, human-centered and resilient European industry [14].

Cobots support production flexibility and efficiency in such a context, favoring human-robot interaction. They can be considered the "ideal new coworker" [15] (p. 2) for their users, who program the robots' motion and collaborate with them in some determined tasks, which are two steps responsible for the user experience (UX) [16].

UX can be defined as the sum of all perceptions, emotions, and responses that users experience when interacting with some technological tool, as well as the ones experienced before and after such an interaction [17]. Therefore, UX derives from the combined result of the expectations prior to the experience, the actual experience during the interaction, and the post-interaction experience [18], trying to holistically understand the humans' side of this relation [19]. Tubin and colleagues [20] advocate that it is necessary to assess UX at different times and use combined methods to fully understand its related aspects.

Hassenzahl [21] affirms that UX integrates both pragmatic and hedonic aspects. On the one hand, the pragmatic or instrumental component of the author's Model of User Experience emphasizes the fulfillment of a behavioral task by an individual, being intrinsically related to the manipulation of the mentioned product. On the other hand, the hedonic component is not focused on the task at hand, but instead on the individual's psychological state. The latter can be related to stimulation that results in personal development, identification with the objects as a way of self-expression, or evocation of valued memories. Designers should aim at the balance between pragmatic and hedonic attributes of UX [22].

UX has become increasingly important on account of the spreading of technology in a society that is shifting from a materialistic to an experiential culture [23], and its importance has been recognized by both researchers and practitioners [24,25]. Given its centrality, a trend towards experience-driven design has arisen. Olsson [26] (p. 165) defined such a design through three assumptions: (a) "takes (user) experience as a starting point; "valuing the whole person behind the 'user'", (b) "uses the targeted experience, and stories around them, as a central concept of the design vision", and (c) "focuses on the key design elements: context, interpretation, participation".

Hassenzahl and Tractinsky [25] clarify that the aim is not to design an experience, but instead to design for an experience. Therefore, first, the intended UX must be defined, and only after that is it possible to come to a decision on how to conjure it [27]. Olsson [26] emphasizes that this decision might benefit from the dialogue between designers and users, to ensure that the perspectives of the latter are taken into consideration.

Throughout this whole design process, UX goals are expected to be identified and utilized [21]. UX goals, which concern the intended experiences that the technology used should provide its users, can be classified as do-goals and be-goals, by its pragmatic or hedonic nature [22]. They can also be designated as instrumental (e.g., ease of use) and non-instrumental (e.g., visual aesthetics) qualities, respectively, as in the Components of User Experience Model developed by Mahlke and Thüring [28], which endorses that both types of characteristics influence the emotional reactions and consequent judgment by users in an interactive context. The instrumental attributes can be considered as staying in an inferior hierarchical position in comparison to the others, and so derive from them e.g., [29]. Some authors even consider that non-task-related goals are the great focus of this kind of design [18,27].

In any case, the goals that users need to be met through the interaction with a cobot or other technological device should be the starting point for the experience-driven design [26]. The priority is to create a pleasurable experience, as the product remains secondary [23]. For the reaching of such experience, UX goals must be clearly defined [27,30]. Such goals must also be precise, measurable, and achievable, though they can be refined and altered throughout the design process [31].

In short, the first step when designing a robot is to formulate the goals intended during the HRC entire process. Thereafter, the necessary technological functionality will be contemplated and, hopefully, materialized [23].

This study aims to meet the need for more in-depth research to explore how cobots are being used and how can they be improved, to guarantee that UX designs successfully fulfill their purpose of creating positive physical and psychological responses. For that purpose, a systematic review of the feasible UX goals for HRC present in the literature of the last eleven years was conducted. Its intention was of enhancing the knowledge of the UX goals as a guide to design cobots, through the contribution of diverse research areas. Besides understanding the UX goals described in the literature, the possible distinct importance attributed to them was also investigated.

Apart from this first introductory section, this paper is structured as follows: Section 2 describes the method used for conducting this systematic literature review; Section 3 presents its results and discusses the descriptive and content analyses; and Section 4 addresses the conclusions of this research.

## 2. Materials and Methods

The present systematic literature review was undertaken following the stages proposed by Donato and Donato [32]. According to these authors, systematic reviews differ from traditional ones in the sense that they are replicable and unbiased. It must be an exhaustive review, to cover all the relevant literature and follow a rigorous methodology.

The first step is to formulate the research questions. Once the research that will be answered in the review is well-established, the inclusion and exclusion criteria must be defined, as well as the search strategy. After that, the papers shall be selected and their quality evaluated. Finally, the data is extracted, synthetically analyzed, and hopefully published. The methodological description of these stages is documented in the following subsections, following the PRISMA framework [33].

The PICo structure (Population or Problem, Interest, Context) [34] was used to define the research questions. Considering the problem related to the application of UX goals in cobot design, in the industrial context, the following research question was defined: Which UX goals should be considered for cobot design in the industrial context? Additionally, to

analyze the relevance of the different UX goals in the cobots design, another question was defined: Do those UX goals have different importance?

## 2.1. Search Strategy Protocol

The search strategy consisted of a comprehensive search that could locate the widest spectrum of articles for consideration and was performed in selected electronic databases, namely, Scopus and Web of Science (Core Collection). These databases were chosen as they are the two main ones used for searching and publishing [35]. The keywords used in this literature review were: 'cobot', 'design', 'user', 'experience', and 'goal'. All these keywords were combined with their synonyms, as can be seen in Table 1.

**Table 1.** Keywords and their synonyms.

| Keywords | Synonyms |
|---|---|
| Cobot | Cobotic<br>Human-robot interaction<br>Human-interactive robot<br>HRI<br>Human-robot collaboration<br>HRC<br>Human-cobot interaction<br>Human-collaborative robot<br>HCI<br>Collaborative robot<br>Collaboration human-robot<br>Human-robot collaborative workstation<br>Co-robotic |
| Design | Plan<br>Delineation<br>Representation<br>Model<br>Proposal<br>Method<br>Framework<br>Experience-driven design<br>EDD<br>User-centered/centred design<br>Human-centered/centred design<br>HCD<br>Design thinking<br>Interaction design<br>Research through design<br>RtD<br>User experience design<br>UXD |
| User | Operator<br>Programmer<br>Human controller<br>Supervisor<br>Facilitator<br>Worker<br>Teammate<br>Human agent |
| Experience | Sense<br>Understanding<br>Perception<br>Usability<br>UX<br>UE<br>Emotion<br>Feeling<br>Event<br>Impression |

**Table 1.** *Cont.*

| Keywords | Synonyms |
|----------|----------|
| Goal | Aim<br>Purpose<br>Objective<br>Target<br>Intention<br>Ambition<br>Requirement<br>Need/necessity<br>Outcome<br>Effect<br>Value<br>Task<br>Accomplishment<br>Safety<br>Trust<br>Fellowship<br>Sympathy<br>Inspiration<br>Satisfying<br>Enjoyable<br>Fun<br>Entertaining<br>Helpful<br>Motivating<br>Aesthetically pleasing<br>Supportive of creativity<br>Rewarding<br>Emotionally fulfilling |

Only studies published between January 2010 and 31 December 2021 were included. From the year 2021, only the publications until the 22 July were considered, hence the search was done on the 23 July.

This time span allows revising the publications that followed the definition of the term 'user experience' by the International Organization for Standardization [17] until the present time. This is especially pertinent given that job opportunities for UX designers are estimated to have increased by 13% since the year 2010 [19].

Posteriorly to the search stage, the retrieval of the results was conducted in two distinct phases. First, they were entered into an electronic spreadsheet and duplicated studies were removed. Then, following the PRISMA framework, the title, abstracts, and keywords of all the remaining papers were read and evaluated [36]. The criteria to transition to the next phase were that the articles needed to include cobots and mention its UX goals. In the cases where the belonging to those criteria was not clear, they transitioned too.

*2.2. Eligibility Criteria*

This review includes literature that was written in English, from a variety of disciplines (e.g., social sciences, robotics, human-robot interaction). Only articles or conference papers were included. The studies that were not developed in an industrial context or that did not fit the theme (i.e., were not actually in the scope of cobots' UX goals) were excluded. The remaining studies were analyzed by reading the full text. Only articles with a focus on the interaction between humans and cobots, that add to the knowledge of UX goals, and referred to the industrial context and that answered the research questions described above were included.

**3. Results and Discussion**

The flow diagram of the PRISMA Statement methodology is presented in Figure 1 [2,22]. A total of 3759 records were obtained (after removing 1537 duplicates). After the application of eligibility criteria for the first phase, 2518 articles were excluded by not being related to cobots and 136 were excluded by not mentioning UX goals. From the 105 articles that

transitioned, all of them were actually developed in an industrial context, so the exclusion factor ended up being the article's allusion to the scope of cobots and UX goals. In the end, seven studies were included to be analyzed in this review.

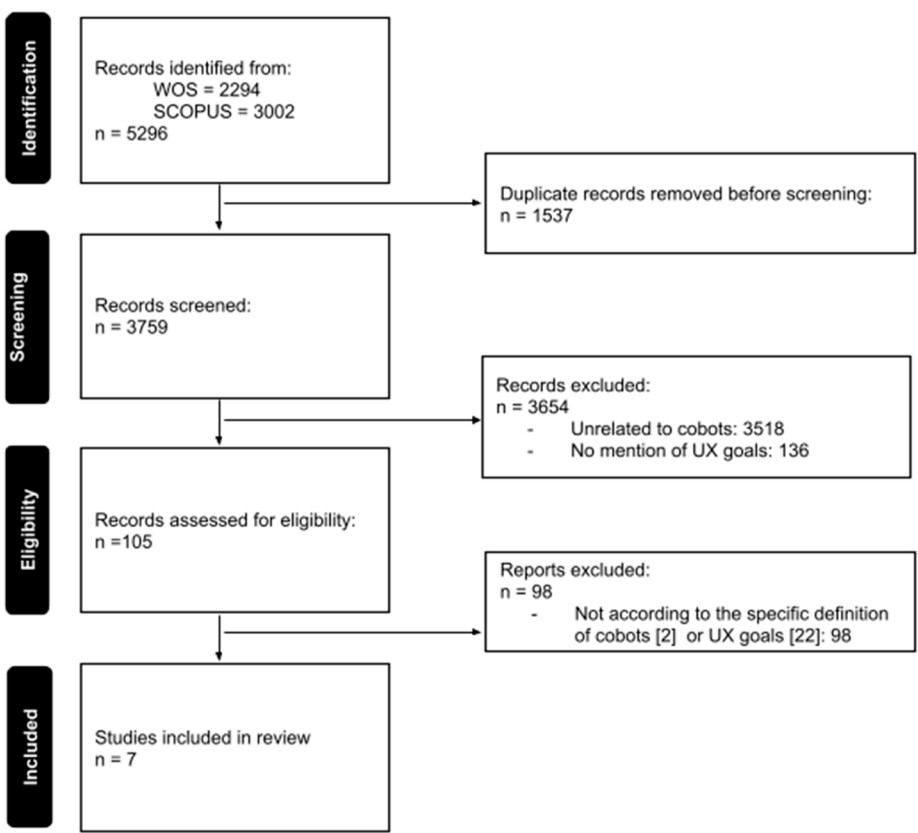

**Figure 1.** PRISMA Statement flow diagram [2,22].

The first selected paper was published in 2015, followed by a two-year gap. After that, there was a slight increase in publications about the topic under study, with four articles being published in 2020 (see Figure 2). There were six conference papers included and one article, all of which provided insights related to how the design of cobots can be enhanced.

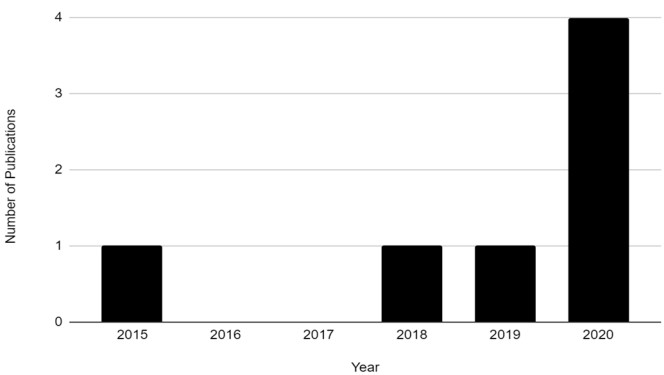

**Figure 2.** Number of publications throughout the years.

Two of the studies were carried out in Finland, and the others in, Spain, Italy, Germany, Turkey, and the United States of America.

These studies focused on a variety of topics, namely industry 4.0 e.g., [37], collaborative robots e.g., [38], human-robot collaboration e.g., [39], user experience, experience-driven design, and UX goals (e.g., [16]). Additionally, other topics were explored, namely, but

not exclusively, task planning [37], barriers and development needs [38], virtual reality, augmented communication, and artificial intelligence [39], technology adoption and social cues [40], robot motion and animation principles [41], and safety [42].

The samples studied were quite diverse. On the one hand, they included students [6,39,41,42], as well as researchers and scientists [16,38]), representing academia. On the other hand, industry was represented by several industrial professionals, from management and supervision [27,40] to different operational roles [6,37,38,40,42], and even end-users [38]. The sample size ranged from a minimum of 10 to a maximum of 140 participants (see Table 2).

**Table 2.** Summary of the studies reviewed.

| Reference | Sample | Research Design | Methods and Instruments | Barrier(s) to the UX Goals Application | Outcomes |
|---|---|---|---|---|---|
| [38] | 75 members of The Robotics Society | Quantitative | Webropol survey platform; online questionnaire | The most significant barrier was the lack of knowledge of, for example, potential applications, reference cases, safety legislation, and ease-of-use. | The most significant development needs were about new ways of allocating work between human workers and cobots, and safety technology. |
| [39] | 80 students | Qualitative | Content analysis | - | The benefits identified across all conditions were combined into the categories of efficiency, assistance, and relationship. |
| [16] | 22 millennials | Mixed method | Observations; semi-structured interviews, short version of User Experience Questionnaire | - | Four user experience goals were identified, namely fellowship and sympathy, inspiration, safety and trust, and accomplishment. |
| [42] | 140 participants | Qualitative | Hands-on demonstration; questionnaires | The main barriers identified included safety, cost, workers' acceptance, and lack of knowledge. Some features expected in a cobot were a universal programming language, programming by demonstration, modularity, and safety features. | The main requirements were considered to be safety, usability, flexibility, and efficiency. |
| [37] | 10 employees at Schaeffler Group | Mixed method | Pick-and-place palletization task; Likert-scale questionnaires | - | The developed system ensured flexibility and comfort, enabling a fluent human-robot collaboration. |
| [40] | 17 manufacturing workers | Qualitative | Observations; semi-structured interviews | - | The themes that emerged from the analysis can be grouped into two key implications for the design of cobots, namely the importance of sociality and the need to support relationships with several stakeholders. |
| [41] | 72 students | Mixed method | Questionnaires; semi-structured interviews | - | The principles of appeal, secondary action, and arcing had a significant positive effect on most outcomes, improving robot perceptions and user experience. |

Regarding the study design, one paper used quantitative methods [38], three used qualitative ones [39,40,42], and three followed a mixed-method approach [6,37,41]. Data collection was conducted in various ways, the most frequent ones being questionnaires and semi-structured interviews (see Table 2).

Five of the seven studies performed experiences with actual cobots, two of which were conducted in real industrial settings (see Table 2). Another study used a virtual reality setting to perform its experiments. Some of the robotic devices were the Kuka robot [39], the Franka Panda cobot ([16]), the Universal Robot [37,41], and the Baxter robot [40].

The experience of users is increasingly being considered when designing technological devices [26], highlighting a human-centric perspective, which is central to the industry 5.0 paradigm [14]. This experience-driven design is enabled by the definition and application of UX goals, that guide the whole process [21]. In this systematic literature review, we

intended to get to know which were the UX goals present in the work published since 2010 that could contribute to an enhanced design of industrial cobots. Table 3 shows how we proceeded to analyze and synthesize the data extracted from the seven selected studies, through its categorization.

**Table 3.** UX goals for the designing of cobots derived from the systematic literature review.

| Reference | Safety | Relationship | Usability | Inspiration | Flexibility | Efficiency | Accomplishment | Score (Study) |
|---|---|---|---|---|---|---|---|---|
| [38] | + | 0 | + | 0 | 0 | + | 0 | 3 |
| [39] | + | + | 0 | 0 | 0 | + | 0 | 3 |
| [16] | + | + | 0 | + | 0 | 0 | + | 4 |
| [42] | + | 0 | + | 0 | + | + | 0 | 4 |
| [37] | + | 0 | + | + | + | 0 | + | 5 |
| [40] | 0 | + | 0 | 0 | 0 | 0 | 0 | 1 |
| [41] | 0 | + | + | + | 0 | 0 | 0 | 3 |
| Score (item) | 5 | 4 | 4 | 3 | 3 | 3 | 2 | |

*Safety* was mentioned by Aaltonen and Salmi [38] when they considered safety technologies, design methods for safety and hygiene requirements as needs for the development of cobots, contributing to their acceptance by the users and consequent successful deployment in industry. Arntz et al. [39] identified assistance as a positive aspect, which can be considered under the safety category when the robotic device assists the human in the least safe tasks, their collaboration benefiting aspects such as work efficiency and alleviation of working conditions. Murali et al. [37] highlighted the importance of the feature of comfort, in the sense of feeling safe, for the implementation of a paradigm with both efficiency and ergonomy at its core. Chowdhury et al. [16] considered the UX goal of safety and trust, as well as Kildal et al. [42], who identified safety as a requirement in this context in which workers and cobots collaborate, sharing a task and workspace, and having targets to achieve.

*Relationship* was identified as a positive element of the experiences run by Arntz et al. [39], specifically through adequate features that assure acceptable communication between a user and a cobot. Chowdhury et al. [16] further emphasized relationship as a UX goal, by mentioning fellowship and sympathy as means to create a bond between both parts. Sauppé and Mutlu [40] pointed out the social aspects of a relationship and the need to develop multiple relationships as implications to be considered when designing cobots, to enrich the social environment of the organizations and provide the required social cues for coordinated actions. Terzioglu et al. [41] referred to the principle of secondary action, which does not contribute to a defined purpose but adds to the lifelikeness of a cobot, and so to the building of a relationship with it.

The *Usability* category was extensively present in the study by Aaltonen and Salmi [38], which contemplated development goals, such as mobile robot cells, use of machine vision, utilization of artificial intelligence, new kinds of user interfaces, utilization of other sensors, programming methods, and mobility. Kildal et al. [42] also identified usability as a requirement to consider cobots's adoption and acceptance. When Murali et al. [37] referred to physical effortlessness, that could be linked with usability, as it concern a physical aspect of a task performed in a collaborative setting. Following the same logic, arcing, one of the principles of the study of Terzioglu et al. [41], can be seen as linked to this category, by addressing the trajectory of the interaction.

*Inspiration* was defined by Chowdhury et al. [16] as a UX goal that relates to feeling motivated and challenged, making the collaboration as fluent as possible. Murali et al. [37] mentioned the feature of mental effortlessness, which can contribute to this goal of inspiration. Terzioglu et al. [41] studied how making a cobot more appealing, manipulating its characteristics such as physical appearance, posture and gaze, can elicit interest and engagement with it, i.e., inspiration.

The UX goal of *Flexibility* was acknowledged by both Kildal et al. [42] and Murali et al. [37], which can be leveraged by the human-robot collaboration models. Arntz et al. [39] pointed out the flexibility of material handling devices as a development need, which also integrates the present category.

In the *Efficiency* category, Aaltonen and Salmi [38] enumerated the following need for cobots development: new ways of allocating work between human workers and cobots; developing performance; and comprehensive solutions taking advantage of the best of robots and humans. Similarly, Arntz et al. [39] and Kildal et al. [42] identified this goal in their research, which could refer, for instance, to increasing precision and speed.

Chowdhury et al. [16] nominated the feeling of success derived from HRC as the UX goal Accomplishment, integrating aspects as confidence, enthusiasm, and pride. The study from Murali et al. [37] also refers to this goal as one of its features, by evaluating the perceived success of an HRC architecture.

To summarize, the answer to our research question regarding the UX goals that can be used for the design of cobots in industry is that seven feasible goals were mapped in the literature of the last eleven years. Such goals are (a) safety, (b) relationship, (c) usability, (d) inspiration, (e) flexibility, (f) efficiency, and (g) accomplishment.

By this categorization, it was possible to observe that not all the goals mentioned before were addressed the same number of times. The number of articles in which they appear can be understood as a measure of their importance within the reviewed literature. This way, safety is ranked as the most relevant, being mentioned in five of the seven articles reviewed. It is followed by the UX goals of relationship and usability, which were both mentioned four time. Then come inspiration, flexibility, and efficiency, with three mentions each. Lastly, accomplishment was phrased in two of the selected articles, being the least relevant of this literature review.

If we try to make the link between the seven UX goals and the distinction between pragmatic and hedonic goals [22], usability, efficiency, and flexibility would be do-goals because of relating to more instrumental aspects of HRC; whereas relationship, inspiration, and accomplishment would be understood as be-goals for concerning non-instrumental aspects. Safety could possibly be perceived as both a pragmatic and hedonic goal, once it comprises task-related aspects (e.g., assistance), but also aspects that relate to the individual psychological state (e.g., comfort). Thereafter, there seems to be a balance between these two types of attributes among the UX goals found in the reviewed studies.

It is possible to aggregate the studies by research design, to understand which clusters elicited which of the seven UX goals. Beginning with the quantitative approach [38], the single study in which this design was used mentioned safety, usability, and efficiency goals. The studies composing the qualitative cluster [7,39,42], referred to the UX goals of safety, relationship, usability, flexibility, and efficiency. Finally, the cluster constituted by the mixed-method articles [16,37,41] included all the goals except for efficiency.

In terms of the pragmatic or hedonic nature of the UX goals, it was observed that all the clusters added to the safety goal, that can be understood as both pragmatic and hedonic. They all also evoked other pragmatic goals. However, only the studies with qualitative and mixed-method designs contributed to the hedonic ones. It was also noted that the inspiration and accomplishment goals were only cited in papers from the mixed-method cluster.

Interestingly, even though we included papers published since 2010 in our search, only papers after the year 2015 were considered eligible for the analysis, most of which were published from 2018 onwards. This might indicate that the use of UX goals as a guide for the designing process of collaborative robots started gaining more relevance only quite recently, especially in the context of industry 5.0, which focuses on the value of technology for the wellbeing of workers [14].

## 4. Conclusions

As this is a recent topic, we suggest that a systematic literature review similar to the present one is done in some years' time, in order to check if the reported UX goals for

the designing of cobots are still being used, if new ones have arisen, and their relative importance. Future research could also include more and different databases in its search, since this review only comprised publications of two databases, which could pose as a limitation of our results. These two limitations may hinder the results to be generalized, given the low number of studies involved, and the lack of countries' representativeness. In addition, it is necessary to consider the new generations of digital technology that can model UX and contribute to the design of cobots, such as digital twin technology [5]. Leng and colleagues [5] studied this type of technology that could be used to support the design of the human-centric engineering system.

Another limitation relates to the fact that not all the authors referenced in this review addressed the UX goals as such. Therefore, some of them were inferred as UX goals, given their description, even though this is a procedure that has the inevitable risk of biases.

However, this said limitation can also be seen as a theoretical implication of our work, despite its inherent subjectivity. To our best knowledge, no attempt to systematically review UX goals has been made, which means this is a pioneering study in that sense. This is the first categorization of the different UX goals that can be utilized for the designing of cobots in a manufacturing setting, and the determined categories can be of use for further research on this topic. One example of how these categories can be applied is by the development of a single questionnaire for the evaluation of these seven UX goals specifically.

Regarding the practical implications of this first study, the most evident one is that it is possible to empirically test if actual cobots match the seven UX goals categorized. That can be done, for instance, through the application of questionnaires and conducting interviews. Another important contribution is that these goals can and must be used at the designing stage of collaborative robotic devices, to ensure that they will comply with the desired UX from the start.

**Author Contributions:** Conceptualization, A.P. and C.C.; methodology, I.M.D.; software, I.M.D.; validation, A.P., C.C. and J.S.; formal analysis, I.M.D.; investigation, I.M.D.; resources, A.P., C.C. and J.S.; data curation, I.M.D.; writing—original draft preparation, I.M.D.; writing—review and editing, I.M.D., A.P., C.C., A.Z. and J.S.; visualization, I.M.D.; supervision, A.P., C.C. and A.Z.; project administration, A.P. and C.C.; funding acquisition, A.P. and C.C. All authors have read and agreed to the published version of the manuscript.

**Funding:** This research was funded by national funds through FCT—Fundação para a Ciência e a Tecnologia, I.P., grant number UIDB/05037/2020.

**Conflicts of Interest:** The authors declare no conflict of interest.

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
