# Peer review of "The Contribution of the User Experiences Goals for Designing Better Cobots: A Systematic Literature Review"

_asi, doi:10.3390/asi5060119_

Round 1

Reviewer 1 Report

1. 160. electronic spreadsheet. spreadsheet is good enough

2. Is going through title, abstract, key words a solid way to exclude 3654 records out of 3759 records? I am a little surprised there are no articles from Japan, China, Korea, etc, where cobots are made or used extensively.

3. 312. The second limitation is that only two data bases are used. Not very clear what is the first one.

4. All in all, the abstract description of seven UX design goals are summarized  in the article. Can the author give a few concrete examples that these goals assist in improving cobot design?

Author Response

Many thanks for reviewing our manuscript and providing insightful suggestions. We have carefully made the following corrections, based on the comments. These changes were marked in the original paper with a different color. The English language was further checked by a native speaker who also teaches the language.

Point 1: 160. electronic spreadsheet. spreadsheet is good enough.

Response 1: Thank you for your appreciation and comments on the article. We will take them into consideration and we will try to improve the article based on your appreciation.

Point 2: Is going through title, abstract, key words a solid way to exclude 3654 records out of 3759 records? I am a little surprised there are no articles from Japan, China, Korea, etc, where cobots are made or used extensively.

Response 2: This is a very relevant question that we also questioned ourselves while conducting this systematic literature review. There was indeed a great amount of articles excluded from one phase to the other, by analyzing the title, abstract and keywords of each article. Nonetheless, we proceeded with the review this way, given that we were following the PRISMA framework. We added more literature to the text (p. 6, ll. 173-175) and improved the definition of the criteria used throughout this process in its different phases (pp. 6, ll., 176-187), besides improving the PRISMA diagram (p. 6, l. 196). Regarding the lack of articles from countries where there is an increasing use of cobots, that is a limitation of our search, which we believe could be tackled in future studies, as added to the article (p. 14, ll. 351-352). However, we did add literature from those countries too (e.g., p. 2 , ll 63-70.).

Point 3: 312. The second limitation is that only two data bases are used. Not very clear what is the first one.

Response 3: Indeed we only used two databases, Scopus and Web of Science Core Collection (which includes the databases Science Citation Index Expanded, Social Science Citation Index, Arts & Humanities Citation Index, and Emerging Sources Citation Index). The decision to choose these two was due to them being considered the two main databases both for searching and for publishing, information that we added in the article (p. 4, ll. 158-159). We believe it would, of course, be relevant to replicate this search in other databases, which we stated as a suggestion for future studies (p. 14, ll. 348-350).

Point 4: All in all, the abstract description of seven UX design goals are summarized in the article. Can the author give a few concrete examples that these goals assist in improving cobot design?

Response 4: Yes, of course. Some examples of the ways in which the seven identified UX goals can contribute to an improved design of cobot were added throughout the text, in the Discussion (p. 13, ll. 253-304). Thank you for this suggestion, we fiercely believe these additions have complemented the description of each UX goal, accentuating their relevance.

Reviewer 2 Report

Overall, this is a well written paper with an interesting result on designing 2 better cobots

INTRODUCTION

The introduction provides sufficient background information for readers to understand the problem, however the authors should clarify the importance of cobots in actual society.

The objectives are clearly defined at the Introduction, the argumentation in this last part was concise and clarifying.

METHODS

The procedure is well explained and is easily comprehensive.

RESULTS

Results include more relevant and extended data.

DISCUSSION

All possible interpretations of the data considered are consistent, however there are some grammar mistakes that should be corrected.

Include limitations and practical applications.

The conclusions have coherence with the initial aims, in addition, they are well established and according to the present discussion. 

Author Response

Many thanks for reviewing our manuscript and providing insightful suggestions. We have carefully made the following corrections, based on the comments. These changes were marked in the original paper with a different color. The English language was further checked by a native speaker who also teaches the language.

Point 1: Overall, this is a well written paper with an interesting result on designing 2 better cobots.

Response 1: Thank you for your appreciation and comments on the article.Point 2: The introduction provides sufficient background information for readers to understand the problem, however the authors should clarify the importance of cobots in actual society. The objectives are clearly defined at the Introduction, the argumentation in this last part was concise and clarifying.

Response 2: Thank you once more for this appreciation. We agree that the importance of cobots in the actual society, namely in the context of industry 4.0 and 5.0, could be further explored, and so we added some literature on that matter (p. 2, ll. 63-70).

Point 3: The procedure is well explained and is easily comprehensive.

 Response 3: Thank you for the positive feedback.

Point 4: Results include more relevant and extended data.

Response 4: We considered this point very useful for us to improve the way the results are presented and discussed. So, we better clarified the seven UX goals, by adding some examples from the reviewed articles throughout the Discussion (p. 13, ll. 253-304).

Point 5: All possible interpretations of the data considered are consistent, however there are some grammar mistakes that should be corrected. Include limitations and practical applications.  The conclusions have coherence with the initial aims, in addition, they are well established and according to the present discussion.

Response 5: We appreciate your comment on the consistency and the coherence of the information on the article, it is crucial for us to have this validation. Regarding the grammar, we reviewed all the text once again and corrected or reformulated the mistakes we found. Afterwards, the manuscript was also checked by an English native speaker, who made more corrections to it.

Reviewer 3 Report

This paper reviewed the contribution of the user experiences goals for designing better cobots. The presentation is satisfactory. A minor revision is suggested for this paper.

Collaborative intelligence is an important feature of robots in Industry 5.0 context. For instance, industry 5.0: prospect and retrospect; secure blockchain middleware for decentralized iiot towards industry 5.0: a review of architecture, enablers, challenges, and directions. The authors may give more discussions and reviews.

My second suggestion is that the author may give more literature view discussions on the new generation of information technology used in capturing and modeling UX in designing cobots. Digital twin technology, which is identified as the next wave of simulation area, could be used to support the design of the engineering system. The digital twin directly conducts validation and test that can quickly locate the malfunction and inefficiency reason, rule out the mistakes, and test the practicability and safety/security of physical solution in execution. For instance, digital twins-based smart manufacturing system design in industry 4.0: a review; digital twins-based remote semi-physical commissioning of flow-type smart manufacturing systems; digital twins-based flexible operating of open architecture production line for individualized manufacturing. The authors may give more discussions and reviews on this issue.

Please enhance the discussions on future research directions.

Author Response

Many thanks for reviewing our manuscript and providing insightful suggestions. We have carefully made the following corrections, based on the comments. These changes were marked in the original paper with a different color. The English language was further checked by a native speaker who also teaches the language.

Point 1: This paper reviewed the contribution of the user experiences goals for designing better cobots. The presentation is satisfactory. A minor revision is suggested for this paper.

Response 1: Thank you for your appreciation and comments on the article. We will take them into consideration and we will try to improve the article based on your appreciation.

Point 2: Collaborative intelligence is an important feature of robots in Industry 5.0 context. For instance, industry 5.0: prospect and retrospect; secure blockchain middleware for decentralized iiot towards industry 5.0: a review of architecture, enablers, challenges, and directions. The authors may give more discussions and reviews.

Response 2: We agree with the importance of highlighting the human factor in the context of Industry 5.0. Based on your comment, we reinforced that both in the Introduction (p. 2, ll. 63-70) and in the Discussion (p. 11, ll. 239-240; p. 14, ll. 342-343).

Point 3: My second suggestion is that the author may give more literature view discussions on the new generation of information technology used in capturing and modeling UX in designing cobots. Digital twin technology, which is identified as the next wave of simulation area, could be used to support the design of the engineering system. The digital twin directly conducts validation and test that can quickly locate the malfunction and inefficiency reason, rule out the mistakes, and test the practicability and safety/security of physical solution in execution. For instance, digital twins-based smart manufacturing system design in industry 4.0: a review; digital twins-based remote semi-physical commissioning of flow-type smart manufacturing systems; digital twins-based flexible operating of open architecture production line for individualized manufacturing. The authors may give more discussions and reviews on this issue.

Response 3: This is a very interesting perspective that we have not considered before. Thank you for bringing up this topic. We have read some literature on the issue and we definitely agree that it is a theme worth of further investigation, reason for which we emphasized it as a future research direction (p.14, ll. 352-355).

Point 4: Please enhance the discussions on future research directions.

Response 4: Indicating future research directions is a crucial aspect of every study, as there is always room for improvement and to acquire new and updated knowledge. In the Conclusion, we have previously specified the suggestion to recreate a similar systematic literature review, including more databases (p. 14, ll. 348-350). Now, we added a number of suggestions for future studies, namely trying to broaden the countries represented in such a review, as well as interpreting the search regarding cobots considering the new generations of digital technology  (p.14, ll. 352-355).

Round 2

Reviewer 3 Report

All my concerns have been well addressed.